# GDaT: GENERALIZABLE DENSITY-AWARE TRANS-FORMER FOR SOLVING THE TRAVELING SALESMAN PROBLEM

## ABSTRACT

Recently, Neural Combinatorial Optimization (NCO) solvers have demonstrated significant potential in solving the Traveling Salesman Problem (TSP). However, existing NCO solvers typically model only the positional features of nodes, neglecting the differences in regional density among the unvisited nodes during route construction. This would hinder their generalization capability on tasks with unseen distributions and varying scales. To address this issue, we propose the **G**eneralizable **D**ensity-**a**ware **T**ransformer (**GDaT**) for solving the TSP. Specifically, GDaT mainly includes two modules: the multi-scale density extraction module and the density-aware attention module. The former generates multiple nested subgraphs of each unvisited node via the k-nearest neighbors strategy and estimates its densities using Gaussian kernels under each nested subgraph. These densities are then fused by a multi-layer perceptron for capturing multi-scale density features for each unvisited node during route construction. The latter leverages the extracted multi-scale density features to guide the attention-based modeling of positional features, enabling the model to perceive variations in problem scale and node distribution, thereby facilitating more accurate next-node selection under unseen distributions and varying scales. Experimental results on synthetic and real-world TSP datasets across diverse scales and distributions demonstrate that GDaT achieves superior generalization performance. The code is available at https://anonymous.4open.science/r/GDaT-31F2.

## 1 INTRODUCTION

The Traveling Salesman Problem (TSP) is a class of NP-hard combinatorial optimization problems with broad applications in logistics scheduling (Konstantakopoulos et al., 2022), electronic design automation (Alkaya & Duman, 2013), and related domains. Traditional methods, such as exact solvers using branch-and-cut algorithms (Applegate et al., 2009) and heuristic approaches based on local search (Helsgaun, 2017) or hybrid metaheuristics (Vidal, 2022), can produce optimal or high-quality near-optimal solutions. These methods require extensive expert-driven design and parameter tuning, while deteriorates rapidly as problem size grows, which limits their scalability and real-world applicability.

In recent years, constructive Neural Combinatorial Optimization (NCO) approaches have received increasing attention due to their fast inference speed and high solving efficiency (Bengio et al., 2021). These methods (Bello et al., 2017; Nazari et al., 2018; Kool et al., 2019) leverage deep neural networks to directly learn solution-construction strategies from data, thereby avoiding the reliance on hand-crafted design in traditional algorithms and significantly reducing development costs. On small-scale TSP instances with specific data distributions, they have demonstrated performance comparable to traditional solvers (Kwon et al., 2020; Hottung et al., 2022; Sun et al., 2024). In particular, they show strong potential in uniformly distributed settings with no more than 100 nodes.

For constructive NCO solvers, they are usually trained on small-scale instances under fixed distributions, such as uniformly distributed problems with 100 nodes, the learned solution construction strategies often fail to generalize to larger scales or unseen distributions. Although some attempts have been made to address the generalization issues of constructive NCO methods, most of them fo-

cus solely on either cross-scale generalization (Luo et al., 2023; Ye et al., 2024; Zheng et al., 2024; Zhou et al., 2025; Luo et al., 2025) or cross-distribution generalization (Zhang et al., 2022; Bi et al., 2022; Fei Liu & Yuan, 2024). Since real-world TSP instances (e.g., TSPLIB95 (Reinelt, 1991)) usually simultaneously exhibit multi-scale and multi-distribution characteristics, jointly considering both scale and distribution is more practically meaningful. Recently, some works have addressed this issue. For example, some studies (Zhou et al., 2023; Liu et al., 2025) achieve omni-generalization (i.e., generalization across diverse scales and distributions) by designing training-level generalization strategies, but relying solely on training strategies may still lead to poor generalization when applied to problems with distributions that is significantly different from the training set. Other works (Gao et al., 2024; Fang et al., 2024) focus on designing local decision mechanisms that are insensitive to scale and distribution.

However, these mechanisms tend to prioritize nearby nodes, which limits their effectiveness in scenarios where selecting distant nodes crucial for minimizing total tour length is required, such as in clustered settings. Therefore, introducing a sense mechanism that enables the model to jointly perceive variations in both scale and distribution is a more general and effective approach. We observe that as the tour is incrementally constructed, the density of the unvisited nodes changes dynamically, driven by variations in both the scale and distribution of the remaining subgraph (see Figure 1). Furthermore, neighborhoods of different sizes around the same node may exhibit distinct distributional patterns (see Figure 2(A)). Capturing such multi-scale density dynamics can reveal the current structural properties of the unvisited nodes and thereby enables the model to jointly perceive variations in both scale and distribution.

Thus, we propose the **G**eneralizable **D**ensity-**a**ware **T**ransformer (**GDaT**), which mainly includes two modules: multi-scale density extraction module and density-aware attention module. The multi-scale density extraction module first selects neighbor sets of varying sizes for each unvisited node using the k-nearest neighbors strategy, forming a series of nested local subgraphs. Then, it applies Gaussian kernel density estimation to each subgraph, where the smooth distance-decay property of the Gaussian kernel allows a more accurate characterization of the influence of nearby nodes on local density. Finally, it fuses the density estimates from different scales using the multi-layer perceptron to obtain a comprehensive representation of the local density for each unvisited node. The density-aware attention module incorporates density information into the query and key vectors via linear summation with positional features. This enables the attention mechanism to adaptively adjust weight allocation based on the multi-scale density features of each node, thereby allowing the model to perceive local structural variations in scale and distribution. This perception facilitates finer discrimination among proximal nodes in dense regions while maintaining sensitivity to distant nodes in sparse regions, ultimately leading to more accurate next-node selection under unseen distributions and varying problem scales.

The contributions of this paper are summarized as follows: (1) This paper proposes **G**eneralizable **D**ensity-**a**ware **T**ransformer (**GDaT**) for solving the TSP. To the best of our knowledge, this is the first NCO constructive approach that explicitly extract node density features to address the challenge of omni-generalization. (2) This paper introduces a multi-scale density extraction module constructs nested subgraphs at multiple scales for each unvisited node during route construction, capturing a comprehensive multi-scale density representation that reveals the local structural characteristics of the unvisited node set. (3) This paper develops a density-aware attention module that integrates multi-scale density features of each unvisited node into positional attention modeling, achieving more accurate next-node selection on tasks with unseen distributions and varying scales. (4) This paper conducts extensive experiments on synthetic and real-world TSP datasets with varying sizes and distributions. The results demonstrate that GDaT achieves superior omni-generalization performance compared to state-of-the-art methods, particularly on large-scale and and those with complex distributions.

## 2 PRELIMINARIES

### 2.1 TSP DEFINITION

This work focuses on the Euclidean TSP. A TSP instance $S$ can be represented by a complete undirected graph $\mathcal{G} = (\mathcal{V}, \mathcal{E})$, where $\mathcal{V} = \{v_i\}_{i=0}^{n-1}$ denotes the set of $n$ nodes with coordinates

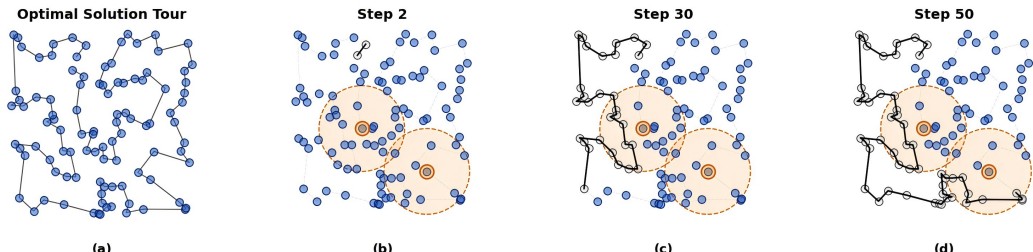

Figure 1: The route construction process. (a) shows the complete solution tour, while (b)-(c) illustrate the local density changes as nodes are sequentially added to the path. Grey nodes represent visited nodes, blue nodes indicate unvisited nodes, and orange regions highlight the local neighborhoods of selected nodes.

$\{\mathbf{c}_i\}_{i=0}^{n-1} \subset \mathbb{R}^2$, and $\mathcal{E} = \{(v_i, v_j) \mid v_i, v_j \in \mathcal{V}, \ i \neq j\}$ denotes the set of edges. A feasible solution can be written as $\boldsymbol{\tau} = (\tau_1, \tau_2, \ldots, \tau_n, \tau_1)$, where $(\tau_1, \tau_2, \ldots, \tau_n)$ is a permutation of the node indices $\{0, 1, \ldots, n-1\}$, and the tour is completed by returning to the starting node $v_{\tau_0}$. The objective is to find a sequence $\boldsymbol{\tau}$ that minimizes the total travel cost, formally expressed as:

$$\min_{\boldsymbol{\tau}} \sum_{k=1}^{n} \left\| \mathbf{c}_{\tau_k} - \mathbf{c}_{\tau_{(k+1) \bmod n}} \right\|_2.$$

## 2.2 Constructive NCO Solvers

Constructive NCO solvers aim to learn end-to-end strategies for solving the TSP. Their core idea is to model the generation of solution sequences as a sequential decision-making process, and most approaches adopt a Transformer-based encoder–decoder architecture (Vaswani et al., 2017). Given a TSP instance with $n$ nodes, the encoder maps the input node features $\{\mathbf{x}_i\}_{i=0}^{n-1} \in \mathbb{R}^{n \times 2}$ into initial node embeddings $\{\mathbf{h}_i\}_{i=0}^{n-1} \in \mathbb{R}^{n \times d}$. The decoder then constructs the complete solution sequence $\boldsymbol{\tau}$ in an autoregressive manner, starting from an initially empty partial solution. At the $t$-th decoding step, the decoder selects a node from the set of unvisited nodes. The selected index $\tau_t$ is then appended to the partial solution $(\tau_1, \ldots, \tau_{t-1})$, with the corresponding node $v_{\tau_t}$ marked as visited, where $\tau_1$ and $\tau_{t-1}$ denote the indices of the first and last visited nodes, respectively. This process repeats until all nodes have been visited, resulting in a complete feasible solution sequence.

## 3 The Proposed GDaT Method

This section describes the proposed GDaT in detail. Firstly, the framework of GDaT is introduced. Then two core modules including the multi-scale density extraction module and the density-aware attention module are presented.

## 3.1 Framework of GDaT

Figure 2 gives the framework of GDaT, which consists of a light encoder and a heavy decoder, and generates solutions in an autoregressive manner. To facilitate the subsequent descriptions, we give the following notations. At the $t$-th step of route construction, let the current partial solution be $\tau_{<t} = (\tau_1, \ldots, \tau_{t-1})$, where $\tau_1$ and $\tau_{t-1}$ denote the indices of the first and last visited nodes, respectively. Denote by $\mathcal{V}_t$ the set of unvisited nodes, and define the context node set as $\mathcal{C}_t = \mathcal{V}_t \cup \{v_{\tau_1}, v_{\tau_{t-1}}\}$.

**Encoder.** The encoder consists of a position feature embedding layer and the proposed multi-scale density extraction module, which are used to encode node position features and multi-scale density features, respectively. Given a TSP instance with $n$ nodes, the input node features $\{\mathbf{x}_i\}_{i=0}^{n-1} \in \mathbb{R}^{n \times 2}$ represent the 2D coordinate vectors of the nodes. The position feature embedding layer transforms these features into initial node embeddings through a linear projection:

$$\mathbf{h}_i^{(0)} = W_{\text{emb}} \mathbf{x}_i + \mathbf{b}_{\text{emb}}, \quad \forall i \in \{0, \ldots, n-1\}, \tag{1}$$

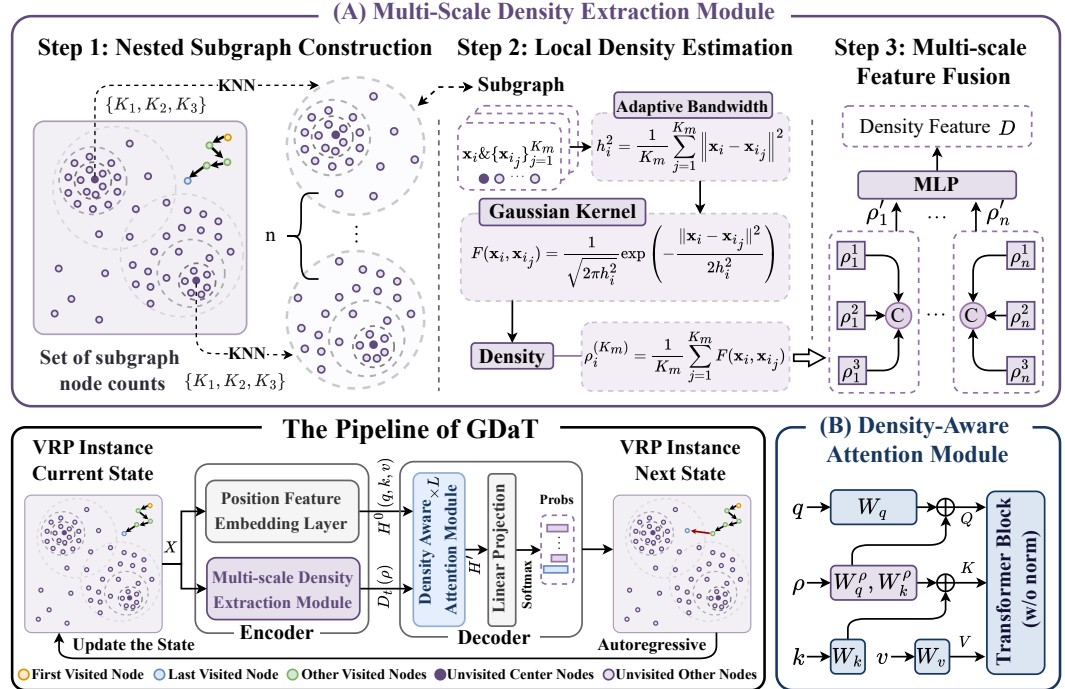

Figure 2: The overall architecture of GDaT.

where $W_{\text{emb}} \in \mathbb{R}^{2 \times d}$ and $\mathbf{b}_{\text{emb}} \in \mathbb{R}^d$ are learnable parameters. The resulting initial position embedding matrix is denoted as $H^0 \in \mathbb{R}^{n \times d}$. This position feature embedding process is performed only once at the beginning of the autoregressive construction. The multi-scale density extraction module computes multi-scale density embeddings $D_t \in \mathbb{R}^{|\mathcal{C}_t| \times d}$ for the current context $\mathcal{C}_t$ at the $t$-th step of route construction (see Section 3.2).

**Decoder.** At the $t$-th decoding step, the decoder initializes its input embeddings by selecting the rows of $H^0$ corresponding to $\mathcal{C}_t$, forming $H_t^{(0)} \in \mathbb{R}^{|\mathcal{C}_t| \times d}$. The embeddings $(H_t^{(0)}, D_t)$ are jointly processed through $L$ layers of the density-aware attention module (see Section 3.3). The output embeddings $H_t^{(L)}$ are used to compute the selection probabilities for the next node via a linear projection followed by softmax. Specifically, the logit for node $v_i$ is computed as:

$$u_i = \begin{cases} \mathbf{h}_{t,i}^{(L)} W_O, & \text{if } i \notin \{\tau_1, \ldots, \tau_{t-1}\}, \\ -\infty, & \text{otherwise}, \end{cases} \qquad \mathbf{p} = \text{softmax}(\mathbf{u}), \tag{2}$$

where $W_O \in \mathbb{R}^{d \times 1}$ is learnable, and $\mathbf{h}_{t,i}^{(L)}$ denotes the row of $H_t^{(L)}$ corresponding to node $v_i \in \mathcal{C}_t$. The next node index $\tau_t$ is sampled from $\mathbf{p}$ and appended to the partial solution. This autoregressive process repeats until a complete solution is obtained.

## 3.2 MULTI-SCALE DENSITY EXTRACTION MODULE

As shown in Figure 2(A), the multi-scale density extraction process consists of three steps: nested subgraph construction, local density estimation, and multi-scale feature fusion. The following describes each step in turn.

**Step 1: Nested subgraph construction.** Neighborhoods of different sizes around the same node may exhibit distinct distributional patterns. To capture these multi-scale views for a more comprehensive representation of local density, we predefine an ordered collection of neighborhood scales $\mathcal{K} = \{k_1, k_2, \ldots, k_m\}$ with $k_1 < k_2 < \cdots < k_m$, where each $k_j$ specifies the number of neighbors for the $j$-th scale.

To avoid redundant neighbor searches, for each node $v_i \in \mathcal{C}_t$, we perform a single $k_m$-nearest neighbors search within $\mathcal{C}_t \setminus \{v_i\}$ to obtain its full neighborhood $\mathcal{N}_i^{(k_m)}$. Smaller-scale neighborhoods

are then derived by truncating this list:

$$\mathcal{N}_i^{(k_j)} = \text{prefix}_{k_j}\left(\mathcal{N}_i^{(k_m)}\right), \quad j = 1, \ldots, m-1, \tag{3}$$

where $\text{prefix}_{k_j}(\cdot)$ denotes taking the first $k_j$ elements. Each neighborhood $\mathcal{N}_i^{(k)}$ defines a nested local subgraph $\mathcal{G}_i^{(k)} = \{v_i\} \cup \mathcal{N}_i^{(k)}$ for all $k \in \mathcal{K}$, which satisfies the nesting property $\mathcal{G}_i^{(k_1)} \subset \mathcal{G}_i^{(k_2)} \subset \cdots \subset \mathcal{G}_i^{(k_m)}$ for all $v_i \in \mathcal{C}_t$. Since we consider a complete undirected graph, the edge set is implicit and thus omitted in our subgraph definition.

**Step 2: Local Density Estimation.** In this work, we compute the density of each nested subgraph using Gaussian kernel density estimation (KDE), which provides a smooth and robust measure of local concentration compared to hard counting methods. For node $v_i$ at scale $k_j$, the density is given by:

$$\rho_i^{(k_j)} = \frac{1}{k_j} \sum_{v_{i'} \in \mathcal{N}_i^{(k_j)}} \frac{1}{h_i^{(k_j)}\sqrt{2\pi}} \exp\left(-\frac{d_{ii'}^2}{2(h_i^{(k_j)})^2}\right), \tag{4}$$

where $\mathbf{c}_i \in \mathbb{R}^2$ denotes the coordinate vector of node $v_i$, $d_{ii'} = \|\mathbf{c}_i - \mathbf{c}_{i'}\|_2$ is the Euclidean distance between nodes $v_i$ and $v_{i'}$, and $h_i^{(k_j)} > 0$ is the bandwidth parameter controlling the smoothness of the estimate.

We employ an adaptive bandwidth strategy in KDE, where the bandwidth for each node is set to the root-mean-square distance to its k-nearest neighbors:

$$h_i^{(k_j)} = \sqrt{\frac{1}{k_j} \sum_{v_{i'} \in \mathcal{N}_i^{(k_j)}} d_{ii'}^2}. \tag{5}$$

This allows the density estimation to automatically adjust to local node distribution, yielding sharper peaks in dense clusters and smoother estimates in sparse regions. Each $\rho_i^{(k_j)}$ thus constitutes a scale-specific and context-dependent estimate of local density, serving as the basis for multi-scale feature fusion in the subsequent step.

**Step 3: Multi-scale Feature Fusion.** For each node $v_i \in \mathcal{C}_t$, stack the scale-specific density estimates into a multi-scale feature vector:

$$\boldsymbol{\rho}_i = \left[\rho_i^{(k_1)}, \ldots, \rho_i^{(k_m)}\right] \in \mathbb{R}^m. \tag{6}$$

This vector is transformed into a learnable embedding by a multi-layer perceptron:

$$\boldsymbol{\rho}_i' = f_{\text{MLP}}(\boldsymbol{\rho}_i), \tag{7}$$

where $f_{\text{MLP}}: \mathbb{R}^m \to \mathbb{R}^d$ is a learnable nonlinear mapping. Stacking $\boldsymbol{\rho}_i'$ for all $v_i \in \mathcal{C}_t$ yields the density embedding matrix:

$$D_t = \begin{bmatrix} (\boldsymbol{\rho}_1')^\top \\ \vdots \\ (\boldsymbol{\rho}_{|\mathcal{C}_t|}')^\top \end{bmatrix} \in \mathbb{R}^{|\mathcal{C}_t| \times d}. \tag{8}$$

The resulting multi-scale density embeddings can reveal the current structural properties of the unvisited nodes and serve to guide the attention-based modeling of positional features.

## 3.3 DENSITY-AWARE ATTENTION MODULE

The density-aware attention module is designed to enhance structural perception by leveraging the extracted multi-scale density features to guide the modeling of positional features. As shown in Figure 2(B), at the $t$-th decoding step, given position embeddings $H_t \in \mathbb{R}^{|\mathcal{C}_t| \times d}$ and multi-scale density embeddings $D_t \in \mathbb{R}^{|\mathcal{C}_t| \times d}$, the attention modeling at the $\ell$-th layer is as follows:

$$\begin{aligned} Q_t^{(\ell)} &= H_t^{(\ell-1)} W_q^{(\ell)} + D_t W_{\rho_1}^{(\ell)}, \\ K_t^{(\ell)} &= H_t^{(\ell-1)} W_k^{(\ell)} + D_t W_{\rho_2}^{(\ell)}, \\ V_t^{(\ell)} &= H_t^{(\ell-1)} W_v^{(\ell)}, \end{aligned} \tag{9}$$

where $W_q^{(\ell)}, W_{\rho_1}^{(\ell)}, W_k^{(\ell)}, W_{\rho_2}^{(\ell)}, W_v^{(\ell)} \in \mathbb{R}^{d \times d}$ are layer-specific learnable matrices. These representations are processed by a Transformer block (without normalization). Specifically, the $\ell$-th block updates node positional embeddings as:

$$
\begin{aligned}
\widehat{H_t}^{(\ell)} &= \mathrm{MHA}(Q_t^{(\ell)}, K_t^{(\ell)}, V_t^{(\ell)}) + H_t^{(\ell-1)}, \\
H_t^{(\ell)} &= \mathrm{FFN}(\widehat{H_t}^{(\ell)}) + \widehat{H_t}^{(\ell)},
\end{aligned}
\tag{10}
$$

where $\mathrm{MHA}(\cdot)$ is a multi-head self-attention layer, and $\mathrm{FFN}(\cdot)$ is a feed forward network with ReLU activation.

Although density embeddings enter Eq. (9) through simple linear addition, the resulting attention scores embody rich interactions. For nodes $v_i$ and $v_j$, the projected query and key at the $\ell$-th layer are:

$$
\begin{aligned}
\mathbf{q}_{t,i}^{(\ell)} &= (W_q^{(\ell)})^\top \mathbf{h}_{t,i}^{(\ell-1)} + (W_{\rho_1}^{(\ell)})^\top \boldsymbol{\rho}_{t,i}, \\
\mathbf{k}_{t,j}^{(\ell)} &= (W_k^{(\ell)})^\top \mathbf{h}_{t,j}^{(\ell-1)} + (W_{\rho_2}^{(\ell)})^\top \boldsymbol{\rho}_{t,j},
\end{aligned}
\tag{11}
$$

so that the unnormalized compatibility expands as:

$$
(\mathbf{q}_{t,i}^{(\ell)})^\top \mathbf{k}_{t,j}^{(\ell)} = \underbrace{\mathbf{h}_{t,i}^{(\ell-1)\top} (W_q^{(\ell)}(W_k^{(\ell)})^\top) \mathbf{h}_{t,j}^{(\ell-1)}}_{\text{pos–pos}} + \underbrace{\mathbf{h}_{t,i}^{(\ell-1)\top} (W_q^{(\ell)}(W_{\rho_2}^{(\ell)})^\top) \boldsymbol{\rho}_{t,j}}_{\text{pos–density}}
$$
$$
+ \underbrace{\boldsymbol{\rho}_{t,i}^\top (W_{\rho_1}^{(\ell)}(W_k^{(\ell)})^\top) \mathbf{h}_{t,j}^{(\ell-1)}}_{\text{density–pos}} + \underbrace{\boldsymbol{\rho}_{t,i}^\top (W_{\rho_1}^{(\ell)}(W_{\rho_2}^{(\ell)})^\top) \boldsymbol{\rho}_{t,j}}_{\text{density–density}}.
\tag{12}
$$

This four-term decomposition demonstrates that positional and density embeddings interact multiplicatively within attention, amplifying differences among proximal nodes in dense regions while enabling sustained exploration of distant nodes in sparse regions, thereby improving next-node selection accuracy under unseen distributions and varying problem scales.

## 4 EXPERIMENTS

This paper conducts comprehensive experiments on both synthetic and real-world TSP datasets, covering a diverse range of problem scales and node distributions, to demonstrate the omni-generalization performance of GDaT. We compare GDaT with several state-of-the-art methods to illustrate its superiority and perform ablation studies to validate the effectiveness of the proposed key components.

**Dataset.** For the synthetic datasets, we generate 16 synthetic TSP datasets by combining four node distributions (uniform, clustered, explosion, implosion) with four scales (1,000, 2,000, 5,000, and 10,000). Each scale-distribution combination contains 128 instances for 1K and 2K, and 16 instances for 5K and 10K. The data generation process follows Fang et al. (2024) . For real-world benchmark datasets, we use 80 symmetric TSP instances from TSPLIB95[1] that provide node coordinates in 2D Euclidean space, with problem sizes ranging from 51 to 33,810 nodes. Additionally, we include 25 symmetric instances from National TSP[2] in World TSP, also given as 2D Euclidean coordinates, with problem sizes ranging from 29 to 24978 nodes.

**Comparison Methods.** We compare our method with: **(1) Traditional Solvers:** LKH3 (Helsgaun, 2017); **(2) NCO Methods for Cross-Scale Generalization:** LEHD (Fu Luo, 2023), GLOP (Ye et al., 2024), UDC (Zheng et al., 2024), DEITSP (Wang et al., 2025a), GELD (Xiao et al., 2025), SIL (Luo et al., 2025), DRHG (Li et al., 2025); **(3) NCO Methods for Omni-Generalization:** Omni-POMO (Zhou et al., 2023), ELG (Gao et al., 2024), INViT (Fang et al., 2024).

**Evaluation Metrics.** We evaluate performance using the average gap to the (near-)optimal solution and the total inference time in seconds (s), minutes (m), and hours (h). For each instance, the gap is computed as:

$$
\mathrm{gap} = \frac{\mathrm{cost}_{\mathrm{model}} - \mathrm{cost}_{\mathrm{opt}}}{\mathrm{cost}_{\mathrm{opt}}} \times 100\%,
\tag{13}
$$

---

[1] http://comopt.ifi.uni-heidelberg.de/software/TSPLIB95

[2] https://www.math.uwaterloo.ca/tsp/world/countries.html

Table 1: Performance comparisons on synthetic TSP datasets of different distributions and problem scales. Symbols denote inference strategies: $G$ denotes single-rollout greedy inference, $G^*$ denotes multi-rollout greedy inference, $A$ denotes the use of data augmentation during inference, $D\&C$ denotes the use of a divide-and-conquer strategy, and $2OPT$ denotes the 2-opt search strategy.

| | Model | TSP-Uniform | | TSP-Clustered | | TSP-Explosion | | TSP-Implosion | | Average gap(%) |
|---|---|---|---|---|---|---|---|---|---|---|
| | | Gap(%) | Time | Gap(%) | Time | Gap(%) | Time | Gap(%) | Time | |
| | LKH3 | 0.00 | 9.8m | 0.00 | 10.4m | 0.00 | 10.3m | 0.00 | 9.7m | 0.00 |
| TSP-1000 | LEHD (NeurIPS'23, $G$) | 2.56 | 1.2m | 15.44 | 1.2m | 6.17 | 1.2m | 4.07 | 1.2m | 7.06 |
| | UDC (NeurIPS'24, $D\&C+G^*+A$) | 2.20 | 42s | 9.16 | 39.5s | 6.44 | 40.3s | 3.22 | 39.8s | 5.26 |
| | GLOP (AAAI'24, $D\&C+G^*+A$) | 4.87 | 15s | 5.30 | 14.2s | 5.17 | 14.3s | 4.65 | 14.2s | 5.00 |
| | ELG (IJCAI'24, $G^*+A$) | 10.53 | 1.3m | 13.29 | 1.2m | 12.67 | 1.2m | 10.67 | 1.2m | 11.79 |
| | INViT-3V (ICML'24, $G^*+A$) | 4.49 | 30.6m | 7.92 | 29.6m | 7.66 | 30.2m | 5.45 | 30.2m | 6.38 |
| | DEITSP (KDD'25, $G+2OPT$) | 3.68 | 3.3m | 5.65 | 3.7m | 4.94 | 3.7m | 4.06 | 3.5m | 4.58 |
| | GELD (arXiv'25, $G$) | 2.76 | 9s | 11.54 | 6s | 5.42 | 5s | 5.80 | 6s | 6.38 |
| | SIL (ICLR'25, $G$) | **1.39** | 20s | 6.92 | 19s | 3.95 | 19s | 3.71 | 19s | 3.99 |
| | GDaT (Ours, $G$) | 2.33 | 5.9m | **2.37** | 4.8m | **2.49** | 5.9m | **2.56** | 5.8m | **2.44** |
| | LKH3 | 0.00 | 44.4m | 0.00 | 41.2m | 0.00 | 43.8m | 0.00 | 46.4m | 0.00 |
| TSP-2000 | LEHD (NeurIPS'23, $G$) | 5.70 | 8.6m | 22.86 | 8.6m | 11.88 | 8.6m | 7.66 | 8.6m | 12.03 |
| | UDC (NeurIPS'24, $D\&C+G^*+A$) | 3.63 | 1.2m | 11.53 | 1.2m | 9.82 | 1.2m | 4.81 | 1.2m | 7.45 |
| | GLOP (AAAI'24, $D\&C+G^*+A$) | 5.67 | 9.5s | 5.97 | 9.5s | 6.08 | 9.6s | 5.51 | 9.6s | 5.81 |
| | ELG (IJCAI'24, $G^*+A$) | 13.44 | 4.7m | 16.01 | 4.7m | 16.63 | 4.7m | 13.38 | 4.7m | 14.87 |
| | INViT-3V (ICML'24, $G^*+A$) | 4.82 | 1.2h | 7.60 | 1.2h | 8.03 | 1.2h | 5.55 | 1.2h | 6.50 |
| | GELD (arXiv'25, $G$) | 4.02 | 11s | 12.80 | 11s | 8.48 | 11s | 5.87 | 10s | 7.79 |
| | SIL (ICLR'25, $G$) | **1.56** | 1.2m | 7.45 | 1.2m | 4.07 | 1.2m | 3.45 | 1.2m | 4.13 |
| | GDaT (Ours, $G$) | 2.62 | 42.7m | **2.72** | 36.2m | **2.75** | 42.3m | **2.65** | 41.7m | **2.69** |
| | LKH3 | 0.00 | 40.0m | 0.00 | 39.6m | 0.00 | 41.4m | 0.00 | 41.7m | 0.00 |
| TSP-5000 | LEHD (NeurIPS'23, $G$) | 14.45 | 15.9m | 35.13 | 15.9m | 20.28 | 15.9m | 15.55 | 15.9m | 21.35 |
| | GLOP (AAAI'24, $D\&C+G^*+A$) | 6.01 | 2.9s | 5.65 | 2.9s | 6.58 | 3.0s | 5.65 | 2.9s | 5.97 |
| | ELG (IJCAI'24, $G^*+A$) | 16.38 | 4.4m | 18.74 | 4.3m | 21.55 | 4.4m | 16.33 | 4.4m | 18.25 |
| | INViT-3V (ICML'24, $G^*+A$) | 5.17 | 21.3m | 6.95 | 21.5m | 8.69 | 21.2m | 5.94 | 21.2m | 6.69 |
| | GELD (arXiv'25, $G$) | 6.98 | 27s | 13.27 | 26s | 11.55 | 27s | 7.90 | 27s | 9.93 |
| | SIL (ICLR'25, $G$) | **2.16** | 1.1m | 7.70 | 1.1m | 6.36 | 1.1m | 3.09 | 1.1m | 4.83 |
| | GDaT (Ours, $G$) | 3.02 | 20.7m | **3.02** | 18.8m | **3.14** | 20.5m | **2.83** | 20.5m | **3.00** |
| | LKH3 | 0.00 | 3.4h | 0.00 | 3.5h | 0.00 | 3.5h | 0.00 | 3.1h | 0.00 |
| TSP-10000 | LEHD (NeurIPS'23, $G$) | 24.86 | 2.1h | 62.36 | 2.1h | 30.36 | 2.1h | 28.66 | 2.1h | 36.56 |
| | GLOP (AAAI'24, $D\&C+G^*+A$) | 5.85 | 9.7s | 4.89 | 9.5s | 5.94 | 9.7s | 5.79 | 9.8s | 5.62 |
| | INViT-3V (ICML'24, $G^*+A$) | 5.19 | 57.1m | 5.77 | 57.8m | 7.30 | 57m | 5.79 | 57m | 6.01 |
| | GELD (arXiv'25, $G$) | 10.08 | 53s | 12.87 | 55s | 12.55 | 54s | 10.02 | 55s | 11.38 |
| | SIL (ICLR'25, $G$) | **2.71** | 3.7m | 9.90 | 3.7m | 5.81 | 3.7m | 3.39 | 3.7m | 5.45 |
| | GDaT (Ours, $G$) | 2.93 | 2.9h | **3.29** | 2.7h | **2.96** | 2.8h | **3.16** | 2.8h | **3.09** |

where $\text{cost}_{\text{model}}$ denotes the tour length produced by the model, and $\text{cost}_{\text{opt}}$ is the optimal or near-optimal tour length. For synthetic datasets, $\text{cost}_{\text{opt}}$ is obtained by LKH3; for real-world TSP instances, $\text{cost}_{\text{opt}}$ is taken from the official best-known solutions.

**Implementing Details.** In the multi-scale density extraction module of GDaT, the number of MLP layers is set to 3, the hidden layer dimension is set to 512, and the number of neighborhood scales per node is set to 3. The choice of scale number and node count per scale is analyzed in Appendix F.1. The density-aware attention module employs multi-head attention with 8 heads, and the hidden dimension of the feed-forward layer is set to 512. In our experiments, this module is integrated within the linear attention framework of Luo et al. (2025), and additional architectural details are provided in Appendix F.2. We adopt a supervised self-improvement training paradigm proposed by Luo et al. (2025) and train on a synthetic dataset of 200,000 TSP-100 instances covering all four distributions (uniform, clustered, explosion, implosion). Training proceeds for 100 epochs with a batch size of 1024. We use the Adam optimizer (Kingma, 2014) with an initial learning rate of $1 \times 10^{-4}$, decayed by 0.97 per epoch. More training details are provided in Appendix G. All experiments are conducted on a single NVIDIA GeForce RTX 3090 GPU with 24GB memory, except for the evaluation on the synthetic dataset (Table 1), which is performed on a single NVIDIA GeForce RTX 4090 GPU with 24GB memory.

## 4.1 COMPARATIVE RESULTS

We conduct extensive experiments on synthetic TSP datasets with four different scales (1K, 2K, 5K, 10K) and four distributions (uniform, clustered, explosion, implosion). To fairly evaluate generalization performance, all competing methods are tested without iterative improvement strategies.

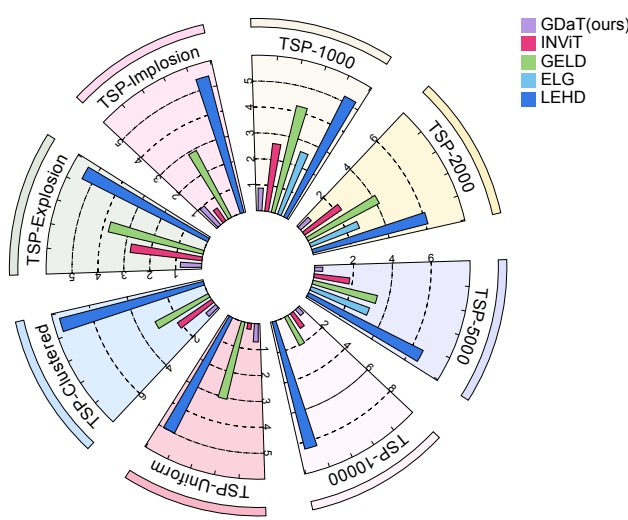

Figure 3: Polar Bar Chart of Variance Analysis on Synthetic Datasets. For better visualization, all variances are shifted by 1 and log-transformed (base 2).

Table 2: Performance comparisons on TSPLIB95 instances grouped by problem size. Each cell shows the average gap and number of successfully solved instances (out of total). *OOM*: The method is inapplicable due to the memory limit. Symbol *I* indicates iterative strategies not directly comparable to ours, hence grouped in Non-iterative. These methods follow the settings of their original papers.

| | Model | 1–100 | 101–1000 | 1001–10000 | >10000 |
|---|---|---|---|---|---|
| **Non-iterative** | LEHD (NeurIPS'23, $G$) | 0.61% (12/12) | 3.28% (37/37) | 13.48% (25/25) | 46.55% (4/6) |
| | UDC (NeurIPS'24, $D\&C+I$) | 0.19% (6/12) | 1.57% (37/37) | 7.54% (19/25) | *OOM* |
| | GLOP (AAAI'24, $D\&C+I$) | **0.32% (12/12)** | **1.01% (37/37)** | 5.64% (25/25) | 6.97% (3/6) |
| | ELG (IJCAI'24, $G^*+A$) | 0.57% (12/12) | 3.93% (37/37) | 11.13% (21/25) | *OOM* |
| | INViT-3V (ICML'24, $G^*+A$) | 1.26% (12/12) | 4.28% (37/37) | 7.76% (25/25) | 8.07% (6/6) |
| | DEITSP (KDD'25, $I$) | 0.67% (12/12) | 1.44% (37/37) | 5.01% (11/25) | *OOM* |
| | SIL (ICLR'25, $G$) | 3.72% (12/12) | 5.62% (37/37) | 6.88% (25/25) | 13.48% (6/6) |
| | GELD (arXiv'2025, $G$) | 0.89% (12/12) | 4.84% (37/37) | 9.23% (25/25) | 15.47% (6/6) |
| | GDaT (Ours, $G$) | 1.00% (12/12) | 2.06% (37/37) | **4.16% (25/25)** | **5.73% (6/6)** |
| **Iterative** | SIL (ICLR'25, T=1000) | 0.28% (12/12) | 0.28% (37/37) | 1.46% (25/25) | 4.57% (6/6) |
| | DRHG (AAAI'25, T=1000) | **0.24% (12/12)** | **0.23% (37/37)** | 2.12% (25/25) | 7.37% (6/6) |
| | GELD (arXiv'25, T=1000) | 0.26% (12/12) | 1.65% (37/37) | 3.83% (25/25) | 6.03% (6/6) |
| | GDaT (Ours, T=(100,1000)) | **0.24% (12/12)** | 0.25% (37/37) | **1.33% (25/25)** | **3.60% (6/6)** |

As shown in Table 1, our method achieves the best performance in 12 out of 16 subsets and consistently outperforms all baselines in terms of the average gap across each scale. Although the uniform distribution remains challenging—where our performance is slightly behind the strongest baseline—our approach still maintains a competitive level. More importantly, Figure 3 presents a variance analysis across eight groups: four by scale (e.g., TSP-1000, aggregating all distributions) and four by distribution (e.g., TSP-Uniform, aggregating all scales), comparing our method with four state-of-the-art generalization-focused models. Our method shows significantly lower variance across all scale groups, demonstrating superior cross-distribution generalization. It also achieves the lowest variance on TSP-Clustered and TSP-Explosion across scales, indicating strong cross-scale robustness in complex distribution settings. Overall, these results indicate that our method achieves state-of-the-art omni-generalization performance without relying on data augmentation or search heuristics.

The performance on TSPLIB95 instances, grouped by problem size, is summarized in Table 2. we divide the 80 TSPLIB95 instances into four groups according to their problem sizes and conduct comprehensive comparisons. Using only greedy inference, GDaT achieves the best performance on

Table 3: Ablation study on Synthetic TSP and TSPLIB95 instances. Results on Synthetic TSP are averaged over all distributions within each scale.

| | Synthetic TSP | | TSPLIB95 | | | |
|---|---|---|---|---|---|---|
| Variant (Ablation) | 1000 | 5000 | 1–100 | 101–1000 | 1001–10000 | >10000 |
| w/o density-aware | 2.75% | 8.20% | 1.66% | 2.27% | 5.74% | 21.60% |
| w/o multi-scale | 3.06% | 6.51% | 1.47% | **1.86%** | 6.00% | 12.84% |
| GDaT (Ours) | **2.44%** | **3.00%** | **1.00%** | 2.06% | **4.16%** | **5.73%** |

the two larger groups and exhibits the smallest performance variation across groups: the gap on the largest group is only 4.73% worse than that on the smallest group, highlighting its robust generalization across both problem scales and the diverse distributions present in real-world instances. We also report results using the iterative improvement strategy employed by SIL (Luo et al., 2025) and GELD (Xiao et al., 2025), and compare them with three state-of-the-art iterative methods. For a fair comparison, all iterative baselines use the number of iterations specified in their original papers. To balance efficiency and solution quality, our method performs 1000 iterations on the two smaller groups, 500 on the 1001–10000 group, and 100 on the largest group. The results show that GDaT with iterative improvement achieves competitive performance on small-scale instances and achieves the best performance on larger scales while using fewer iterations, further demonstrating its superior generalization across scales and distributions. Additional comparative results are provided in Appendix H.

## 4.2 ABLATION STUDY

We conduct ablation studies on synthetic TSP instances of size 1000 and 5000, as well as on real-world TSPLIB95 benchmark, to validate the effectiveness of key components in our method. Specifically, we compare three variants: (1) *w/o density-aware*, which removes both the multi-scale density extraction and density-aware attention modules; (2) *w/o multi-scale*, which retains only the single-scale (largest-scale) density estimation in the density extraction module. All variants are trained using the same protocol for fair comparison.

As shown in Table 3, the incorporation of density modeling (both in single-scale and multi-scale forms) leads to consistently better generalization, particularly on large-scale instances. When comparing *w/o multi-scale* with *w/o density-aware*, we observe that introducing node density estimation, even within a single-scale framework, results in superior overall performance. Although the gains are modest on smaller instances, the improvement becomes significantly more pronounced on larger scales (e.g., TSP-5000 and TSPLIB95 >10K), which indicates that density-aware modeling enhances generalization to large-scale problems. Furthermore, extending the density modeling from a single-scale to a multi-scale design yields additional and substantial performance gains across the board. Our GDaT consistently outperforms the single-scale variant, with the most notable improvements observed in the largest instance groups. This demonstrates that multi-scale density modeling plays a critical role in achieving robust omni-generalization on both synthetic and real-world TSP instances.

## 5 CONCLUSION

This paper proposes the **G**eneralizable **D**ensity-**a**ware **T**ransformer (**GDaT**) for solving the TSP. By extracting multi-scale density features of unvisited nodes and incorporating them into attention-based modeling, GDaT enables more informed and accurate next-node selection under unseen distributions and varying problem scales. Extensive experiments on both synthetic and real-world TSP datasets demonstrate that GDaT outperforms state-of-the-art methods, particularly on large-scale instances and instances with complex node distributions, which highlights its strong omni-generalization performance. A limitation of GDaT lies in the computational overhead incurred by repeatedly computing multi-scale densities during autoregressive route construction. Future work will focus on improving the efficiency of density computation, as well as extending the proposed framework to other combinatorial optimization problems.

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

## A   APPENDIX

## B   ETHICS STATEMENT

This work follows the ICLR Code of Ethics. This study does not involve human subjects or animal experiments. All datasets used, including synthetic TSP datasets, TSPLIB95, and National TSP, were collected in compliance with relevant usage guidelines and do not involve privacy issues. We have taken care to avoid any bias or discriminatory outcomes in our research process. No personally identifiable information was used, and no experiments were conducted that could raise privacy or security concerns. We are committed to maintaining transparency and integrity throughout the research process.

## C   REPRODUCIBILITY STATEMENT

We have made every effort to ensure that the results presented in this paper are reproducible. All code and datasets have been made publicly available in an anonymous repository to facilitate replication and verification. The experimental setup, including training steps, model configurations, and hardware details, is described in detail in the paper. Additionally, the public datasets used in this paper, such as benchmark instances for the Traveling Salesman Problem in combinatorial optimization, including TSPLIB95 and National TSP, are publicly available to ensure consistent and reproducible evaluation results.

## D   LLM USAGE

Large Language Models (LLMs) were used to assist in improving the language clarity and grammatical accuracy of this manuscript. Specifically, the model helped refine sentence structures, correct grammar, and enhance overall readability. We carefully reviewed and edited all text, ensuring that the final content accurately reflects our original ideas and scientific contributions. We confirm that the use of LLMs adheres to ethical guidelines and does not lead to plagiarism or any form of academic misconduct.

## E   RELATED WORK

This section reviews recent NCO methods that aim to improve generalization, which can be broadly categorized into three scenarios: *cross-scale generalization*, *cross-distribution generalization*, and *omni-generalization*.

### E.1   CROSS-SCALE GENERALIZATION

Strategy-based approaches aim to develop generalization techniques that are independent of the specific NCO solver, to handle large-scale instances. Some works focus on designing training algorithms to enhance scalability. Zhou et al. (2024) propose a three-stage variable-scale training scheme to improve cross-scale generalization. Similarly, Luo et al. (2023) introduce a method to learn the construction of partial solutions at varying scales during training for the same purpose. In addition, Luo et al. (2025) propose a self-improvement training paradigm that combines the strengths of reinforcement learning and supervised learning, enabling efficient training on large-scale instances and further improving model adaptability. Other research focuses on designing divide-and-conquer strategies to tackle large-scale problems (Pan et al., 2023; Hou et al., 2023; Ye et al., 2024; Zheng et al., 2024; Zhou et al., 2025). Among them, Ye et al. (2024) and Zheng et al. (2024) leverage Graph Neural Networks (GNNs) to partition large-scale problems into multiple simpler sub-problems, enabling more efficient processing of large instances through parallel sub-problem solving.

From the model-design perspective for cross-scale generalization, Zhou et al. (2024) design an instance-conditioned adaptation module, which explicitly incorporates problem-scale information into the attention mechanism to make the model scale-aware. Similarly, Wang et al. (2025b) propose a Distance-aware Attention Reshaping (DAR) method that leverages scale-related signals to

guide the model's adaptation to varying problem sizes. In contrast, Luo et al. (2023) propose a Light Encoder and Heavy Decoder (LEHD) structure, which enables the model to learn scale-independent features. Additionally, some diffusion-based methods (Sun & Yang, 2023; Zhao et al., 2024) have been found useful for cross-scale generalization.

### E.2 CROSS-DISTRIBUTION GENERALIZATION

Another line of research focuses on improving robustness to unseen distributions. Zhang et al. (2022) present an adaptive curriculum learning strategy based on task difficulty to improve cross-distribution generalization for NCO solvers. Jiang et al. (2022) strengthen generalization by grouping training instances according to their generating distributions and minimizing the worst-case loss across groups. Building on this direction, Bi et al. (2022) propose an adaptive multi-distribution knowledge distillation framework that transfers the strategies of multiple teachers trained on different distributions into a single student model, thereby improving out-of-distribution performance. Overall, these methods mitigate performance degradation in cross-distribution scenarios by optimizing training strategies.

### E.3 OMNI-GENERALIZATION

Since real-world TSP instances often exhibit diverse distributions and varying scales, recent studies have begun to simultaneously address both aspects. From the strategy side, Zhou et al. (2023) propose a general meta-learning framework, while Liu et al. (2025) introduce the Purity Policy Optimization training paradigm, both aiming to simultaneously boost cross-distribution and cross-scale generalization. From the model-design side, another line of research aims at omni-generalization by developing local decision mechanisms that are insensitive to both scale and distribution. For example, Gao et al. (2024) integrate local policies learned from neighborhood information with global policies learned from complete instances, and jointly train them to achieve complementary effects. Fang et al. (2024), on the other hand, restrict the decision space to the neighborhood of the last visited node. Different from previous works, our GDaT focuses on designing modules that are simultaneously scale-aware and distribution-aware.

## F METHOD DETAILS

This section presents the detailed implementation of the proposed GDaT model, focusing on two key modules: the multi-scale density extraction module and the density-aware attention module.

### F.1 SELECTION OF NEIGHBORHOOD SCALES

To determine the appropriate neighborhood scales for the multi-scale density extraction module, we conduct a statistical analysis on the optimal solutions of the entire training set, which consists of 200K TSP-100 instances. Specifically, for each instance, we examine the optimal tour and record, at each step, the nearest-neighbor rank of the next node among all unvisited nodes with respect to the current node. We denote the maximum such rank across all steps in instance $i$ as $k_i$, which reflects the farthest node (in rank) considered by the optimal solution when selecting the next node during the tour construction.

Let $N$ denote the size of the dataset. Given $\{k_i\}_{i=1}^{N}$, we compute:

$$k_{\min} = \frac{1}{N} \sum_{i=1}^{N} k_i, \tag{14}$$

$$k_{\max} = \max_{1 \leq i \leq N} k_i, \tag{15}$$

where $k_{\min}$ reflects the average farthest neighbor considered by the optimal solutions, while $k_{\max}$ corresponds to the worst-case dependency across all instances. In our analysis on the training set, we obtain $k_{\min} = 16$ and $k_{\max} = 98$. To capture richer density-aware features, we further introduce a third scale $k_{\mid}$, defined as the average of the two, yielding $k_{\mid} = 57$.

---

**Algorithm 1** Density-Aware Linear Attention Module

1: **Input:** Node position embeddings $H_t^{(\ell-1)} \in \mathbb{R}^{|\mathcal{C}_t| \times d}$, density embeddings $D_t \in \mathbb{R}^{|\mathcal{C}_t| \times d}$
2: **Output:** Updated node position embeddings $H_t^{(\ell)} \in \mathbb{R}^{|\mathcal{C}_t| \times d}$
3: **Extract representative node position embeddings and density embeddings:**
4: $H_{t,a}^{(\ell-1)} = [H_t^{(\ell-1)}]_a$, $D_{t,a} = [D_t]_a$ {$a$ denotes first and last visited nodes}
5: **Update representative nodenode position embeddings:**
6: $Q_a \leftarrow H_{t,a}^{(\ell-1)} W_q^{(\ell)} + D_{t,a} W_{\rho_1}^{(\ell)}$
7: $K_a \leftarrow H_t^{(\ell-1)} W_k^{(\ell)} + D_t W_{\rho_2}^{(\ell)}$
8: $V_a \leftarrow H_t^{(\ell-1)} W_v^{(\ell)}$
9: $\widehat{H}_{t,a}^{(\ell)} \leftarrow \text{MHA}(Q_a, K_a, V_a) + H_{t,a}^{(\ell-1)}$
10: $H_{t,a}^{(\ell)} \leftarrow \text{FFN}(\widehat{H}_{t,a}^{(\ell)}) + \widehat{H}_{t,a}^{(\ell)}$
11: **Update context node position embeddings:**
12: $Q_c \leftarrow H_t^{(\ell-1)} W_q'^{(\ell)} + D_t W_{\rho_3}^{(\ell)}$
13: $K_c \leftarrow H_{t,a}^{(\ell)} W_k'^{(\ell)} + D_{t,a} W_{\rho_4}^{(\ell)}$
14: $V_c \leftarrow H_{t,a}^{(\ell)} W_v'^{(\ell)}$
15: $\widehat{H}_t^{(\ell)} \leftarrow \text{MHA}(Q_c, K_c, V_c) + H_t^{(\ell-1)}$
16: $H_t^{(\ell)} \leftarrow \text{FFN}(\widehat{H}_t^{(\ell)}) + \widehat{H}_t^{(\ell)}$
17: **return** $H_t^{(\ell)}$

---

Table 4: Ablation study on the number of neighborhood scales.

| Model | 1–100 | 101–1000 | 1001–10000 | >10000 |
|---|---|---|---|---|
| GDaT-2V | 1.25% | 2.52% | 4.49% | 7.28% |
| GDaT-3V | **1.00%** | **2.06%** | **4.16%** | **5.73%** |

To evaluate the impact of varying neighborhood scales, we compare two-scale and three-scale variants on the TSPLIB95 benchmark, as shown in Table 4. GDaT-2V uses $\{k_{\min}, k_{\max}\}$, while GDaT-3V uses $\{k_{\min}, k_{\mid}, k_{\max}\}$. The results demonstrate that adding the third scale consistently improves performance. Given the additional computational cost of further increasing the number of scales, we adopt a fixed three-scale design in all experiments.

### F.2 DENSITY-AWARE LINEAR ATTENTION MODULE

In our experiments, we integrate the proposed density-aware attention module into the linear attention framework of Luo et al. (2025), forming the **Density-Aware Linear Attention Module** as the core component of the decoder. The core idea of the linear attention is as follows: the context node embeddings are first used to update the representative node embeddings (i.e., the first and last visited nodes), and then the updated representative node embeddings are used to update the context node embeddings. This procedure ensures effective information propagation while achieving linear time and space complexity. Algorithm 1 provides the detailed pseudocode of this module.

## G   TRAINING DETAILS

In training neural combinatorial optimization (NCO) models, supervised learning (SL) approaches often struggle to obtain sufficient (near-)optimal solutions as labels, while reinforcement learning (RL) methods suffer from sparse rewards and high GPU memory usage. To address these challenges, Luo et al. (2025) proposed a supervised self-improvement training (SIT) paradigm. The core idea of SIT is to first generate an initial solution, which serves as pseudo-labels for model training. The trained model then reconstructs solutions using a local improvement strategy, producing improved pseudo-labels that are used to further refine the model. By iterating solution reconstruction in this

Table 5: Performance comparisons on synthetic TSP instances under a clustered distribution with 5 clusters. *OOM*: The method is inapplicable due to the memory limit. Symbols denote inference strategies: $G$ denotes single-rollout greedy inference, $G^*$ denotes multi-rollout greedy inference, $A$ denotes the use of data augmentation during inference, $D\&C$ denotes the use of a divide-and-conquer strategy, and $2OPT$ denotes the 2-opt search strategy.

| Model | TSP-1000 | | TSP-2000 | | TSP-5000 | | TSP-10000 | |
|---|---|---|---|---|---|---|---|---|
| | Gap(%) | Time | Gap(%) | Time | Gap(%) | Time | Gap(%) | Time |
| LKH3 | 0.00 | 9.9m | 0.00 | 42.0m | 0.00 | 39.8m | 0.00 | 3.3h |
| LEHD (NeurIPS'23, $G$) | 14.09 | 1.2m | 20.42 | 8.6m | 39.53 | 15.9m | 72.53 | 2.1h |
| UDC (NeurIPS'24, $D\&C+G^*+A$) | 9.47 | 39.6s | 11.95 | 1.2m | *OOM* | | *OOM* | |
| GLOP (AAAI'24, $D\&C+G^*+A$) | 5.19 | 14.23s | 6.18 | 9.58s | 5.81 | 2.9s | 5.25 | 9.39s |
| ELG (IJCAI'24, $G^*+A$) | 12.66 | 1.2m | 15.33 | 4.7m | 18.42 | 4.3m | *OOM* | |
| INViT-3V (ICML'24, $G^*+A$) | 8.58 | 29.5m | 8.59 | 1.2h | 7.95 | 21.7m | 6.76 | 58.3m |
| DEITSP (KDD'25, $G+2OPT$) | 5.93 | 3.6m | *OOM* | | *OOM* | | *OOM* | |
| GELD (arXiv'25, $G$) | 12.58 | 5s | 13.09 | 11s | 14.69 | 27s | 14.02 | 54s |
| SIL (ICLR'25, $G$) | 7.57 | 19s | 7.44 | 1.2m | 7.99 | 1.1m | 9.45 | 3.7m |
| GDaT (Ours, $G$) | **2.20** | 5.9m | **2.62** | 33.4m | **3.49** | 19.2m | **3.27** | 2.5h |

Table 6: Performance comparisons on National TSP instances grouped by problem size. Each cell shows the average gap and number of successfully solved instances (out of total). Symbol $I$ denotes iterative improvement strategies inherent to the respective methods. These approaches follow the settings of their original papers. † indicates results taken from Xiao et al. (2025).

| Model | 1–100 | 101–1000 | 1001–10000 | >10000 |
|---|---|---|---|---|
| Omni-TSP (ICML'23, $G^*$)† | 2.63% (2/2) | 16.02% (4/4) | 79.67% (13/13) | 71.67% (1/6) |
| LEHD (NeurIPS'23, $G$)† | 0.12% (2/2) | 39.94% (4/4) | 82.43% (13/13) | 98.52% (1/6) |
| UDC (NeurIPS'24, $D\&C+I$)† | – | 7.68% (4/4) | 23.21% (13/13) | 18.41% (1/6) |
| GLOP (AAAI'24, $D\&C+I$) | 3.76% (2/2) | 4.55% (4/4) | 6.49% (13/13) | 5.99% (6/6) |
| ELG (IJCAI'24, $G^*+A$)† | 2.28% (2/2) | 11.18% (4/4) | 44.64% (13/13) | 22.44% (1/6) |
| INViT-3V (ICML'24, $G^*+A$)† | **0.03%** (2/2) | 4.94% (4/4) | 10.86% (13/13) | 9.84% (6/6) |
| SIL (ICLR'25, $G$) | **0.03%** (2/2) | 25.73% (4/4) | 59.96% (13/13) | 92.01% (6/6) |
| GELD (arXiv'2025, $G$) | 0.41% (2/2) | 3.96% (4/4) | 17.01% (13/13) | 17.45% (6/6) |
| GDaT (Ours, $G$) | **0.03%** (2/2) | **2.48%** (4/4) | **5.62% (13/13)** | **5.88% (6/6)** |

manner, SIT enables NCO methods to effectively solve large-scale problems without requiring any labeled data.

However, since our model is trained solely on small-scale TSP-100 instances, multiple rounds of self-improvement are not required. Specifically, we first use the LKH3 solver (Helsgaun, 2017) to quickly generate high-quality approximate solutions for the training set as pseudo-labels. The model is then trained for 50 epochs using supervised learning. After a single round of local reconstruction to generate improved pseudo-labels, training continues for another 50 epochs. This single round of self-improvement is sufficient to achieve good performance while substantially reducing training time.

## H  ADDITIONAL RESULTS

We further conduct comparative experiments on (i) a synthetic datasets with a clustered distribution containing five clusters across four problem scales, and (ii) the real-world National TSP benchmark. These additional evaluations aim to verify the robustness and generalization of our method under more complex distributions and real-world instances.

Table 5 reports the results on the synthetic clustered datasets. The results show that, even with greedy inference only, our method consistently achieves the best performance across all scales, demonstrating strong generalization to complex clustered distributions. Table 6 presents the results on the National TSP dataset. Compared with the results on TSPLIB95 (see main text), we observe that most competing methods suffer from significant performance degradation, with the exception

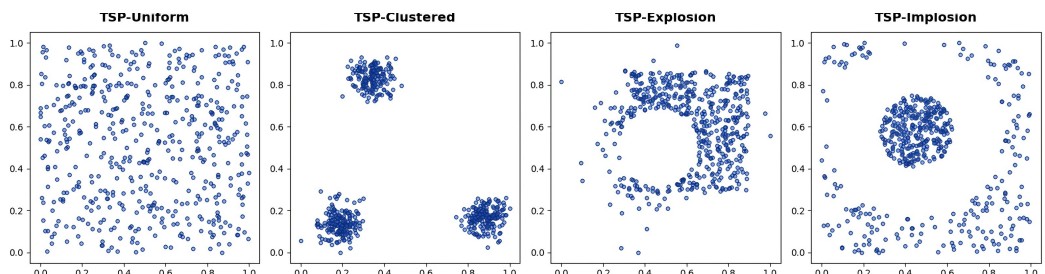

Figure 4: TSP-500 Instances under four different distributions.

of GLOP, INViT, and our GDaT. In particular, GDaT not only maintains stable performance but also achieves the best overall results, which highlights its effectiveness and robustness on real-world benchmarks.

# I    VISUALIZATION

In this work, we consider four distinct node distributions to evaluate the generalization performance of our model. Figure 4 visualizes representative instances of each distribution with $N = 500$ nodes. All instances are normalized to the unit square $[0, 1]^2$ using min-max scaling while preserving aspect ratio. The generation process and characteristics of each distribution are as follows:

**Uniform distribution.** All nodes are independently and uniformly sampled from $[0, 1]^2$:

$$\mathbf{p}_i \sim \mathcal{U}([0, 1]^2), \quad i = 1, \ldots, N, \tag{16}$$

where $\mathbf{p}_i = (x_i, y_i)$. This distribution exhibits uniform node density with no local structure.

**Clustered distribution.** The $N$ nodes are divided into $K$ clusters of approximately equal size. Each cluster center $c_k$ is uniformly sampled from $[0, 1]^2$, and the coordinates of nodes within the cluster are generated by adding isotropic Gaussian noise with standard deviation $\sigma = 0.04$:

$$\mathbf{p}_i = c_k + \epsilon, \quad \epsilon \sim \mathcal{N}(0, \sigma^2 I), \tag{17}$$

where $I$ is the $2 \times 2$ identity matrix. In our experiments, $K = 3$. This results in multiple dense groups separated by sparse regions.

**Explosion distribution.** Nodes are first uniformly sampled from $[0, 1]^2$. Then, for each instance, an explosion center $c \in [0, 1]^2$ and radius $r \sim \mathcal{U}(0.1, 0.5)$ are selected. For any node $\mathbf{p}_i$ within the disc of radius $r$ centered at $c$, its position is updated as:

$$\mathbf{p}'_i = c + \frac{\mathbf{p}_i - c}{\|\mathbf{p}_i - c\|} \cdot \delta, \quad \delta \sim \text{Exp}(\lambda), \ \lambda = 8, \tag{18}$$

pushing it radially outward. This creates a central void with higher node density around the perimeter.

**Implosion distribution.** Nodes are initially sampled uniformly from $[0, 1]^2$. An implosion center $c$ and radius $r \sim \mathcal{U}(0.1, 0.5)$ are sampled. For nodes inside the disc, the position is updated as:

$$\mathbf{p}'_i = c + \alpha \cdot (\mathbf{p}_i - c), \tag{19}$$

where $\alpha = \min(r, |\mathcal{N}(0, 1)|)$ is a scaling factor derived from a half-normal distribution and clipped by $r$. This results in a dense core around $c$ with sparser surroundings.

