# OpenReview forum: "GDaT: Generalizable Density-aware Transformer for Solving the Traveling Salesman Problem"
_ICLR.cc/2026/Conference — Submitted to ICLR 2026_

### Official Review · Reviewer_AdAJ · 2025-10-18

**Soundness:** 3
**Presentation:** 4
**Contribution:** 3
**Rating:** 6
**Confidence:** 4

**Summary:**

This paper proposes the Generalizable Density-aware Transformer (GDaT) to address the "omni-generalization" challenge (cross-scale and cross-distribution) in constructive NCO solvers for the Euclidean TSP. GDaT integrates two core modules: 1) Multi-scale Density Extraction and 2) Density-aware Attention.

**Strengths:**

1. The paper is well-structured and exhibits a polished presentation. The tables, equations, and the visual elements are laid out clearly and elegantly.
2. The integration of Gaussian KDE and density-aware attention is a novel adaptation to TSP's structural dynamics.
3. The proposed GDaT is towards addressing a practical gap (real-world TSP's multi-scale/distribution nature), and can potentially be applied as a more robust backbone for enhancing existing NCO approaches.

**Weaknesses:**

1. The evaluation scope (solely on TSP) is a bit limited. Involving at least similar graph-based CO problems like CVRP, MIS, or simply asymmetric TSP (ATSP) would be better.
2. Omni-generalization is crucial, and so is the in-distribution performance. I suggest the authors report results with the model trained and tested on the same scaled and distributed (uniform) TSP-50/100/500/1k data, following the standard benchmarking test sets adopted by the vast majority of NCO literature (e.g., Att-GCN, GNNGLS, DIMES, POMO, DIFUSCO, etc., where 1280 instances are included for TSP-50/100, and 128 instances for TSP-500), so that a wider applicability and comparability can be ensured for GDaT as a generic TSP backbone.
3. I appreciate the performance of GDaT reported thus far among the constructive NCO family. Yet, reporting comparative results with a broader spectrum of NCO solving paradigms is also important. E.g., comparing your model with the latest heatmap-based generative approaches (e.g., DIFUSCO, Fast-T2T, COExpander), recent sequential solvers (e.g., Sym-NCO, GOAL, BQ-NCO), and more conventional and mainstream baselines (e.g., DIMES, Att-GCN, GNNGLS, etc.), all with greedy decoding policies on the above-mentioned consistent datasets (commonly with small-to-middle sizes). **Note that it's not a problem if GDaT fails to outperform some of these neural solvers on certain test datasets. It's the comprehensive comparison that matters and helps better position a newly developed backbone architecture for future research options.**
4. Only the gap metric is reported in the experimental results. Providing the original objective value (tour length) is recommended for transparent comparison against previous works as well as for easier alignment of future works.

**Questions:**

For clarity, please refer to the Weaknesses section for all my concerns and suggestions. I would like to keep my positive recommendation of this submission contingent on the possible discussions and/or supplementary explanation/results regarding the above mentioned issues.

---

### Official Review · Reviewer_fNZH · 2025-10-21

**Soundness:** 2
**Presentation:** 3
**Contribution:** 2
**Rating:** 4
**Confidence:** 4

**Summary:**

This manuscript proposes GDaT to solve TSP with unseen distributions and varying scales.

**Strengths:**

**S1:**  Although GDaT employs a multi-view approach similar to INViT [1], the authors present a novel model that efficiently processes multi-view information.

**S2:**   This manuscript is well-structured.

[1] INViT: A Generalizable Routing Problem Solver with Invariant Nested View Transformer, ICML 2024.

**Weaknesses:**

**W1:**  This manuscript should evaluate the generalizability of GDaT on other routing problems such as CVRP, as in [1,2]. Evaluating GDaT only on TSP raises concerns about the method’s generality.

**W2:**  This manuscript  lacks comparisons with state-of-the-art methods (e.g., [3]).

**W3:**  GDaT is trained on four distributions, while most baselines use single-distribution (uniform) data. This may bias the comparison.

**W4:**  Inference efficiency is relatively poor. This manuscript doesn't show how performance compares when inference time is similar across models.

**W5:**   Although the authors provided an anonymous link, it appears to be expired; the absence of accessible source code reduces the paper’s reproducibility.

[2] Boosting neural combinatorial optimization for large-scale vehicle routing problems, ICLR 2025.

[3] Learning to Reduce Search Space for Generalizable Neural Routing Solver, arXiv 2503.03137.

**Questions:**

**Q1:** Could the authors clarify the experimental setup for the variance analysis on the synthetic datasets shown in Figure 3? In particular, why do some models exhibit such large variance?

---

### Official Review · Reviewer_qXXJ · 2025-11-01

**Soundness:** 2
**Presentation:** 3
**Contribution:** 2
**Rating:** 4
**Confidence:** 4

**Summary:**

This paper proposes GDaT (Generalizable Density-aware Transformer), a Transformer-based model for the Traveling Salesman Problem (TSP).
The central idea is to enhance model generalization by introducing multi-scale density features that describe local node distributions. Specifically, the authors:

- construct multiple KNN-based subgraphs for each node,

- compute Gaussian kernel density estimates at different scales, and

- inject these density embeddings into the Transformer’s attention mechanism.

Experiments are conducted on synthetic TSP distributions (uniform, clustered, explosion, implosion) and real-world datasets (TSPLIB95, National TSP).  Results show modest improvements over prior Transformer-based solvers such as SIL, INViT, and GELD, particularly in cross-distribution generalization.

**Strengths:**

Reasonable Motivation: The idea of considering local density information is intuitive and relevant for irregular node distributions.

Clear Implementation: The method is technically understandable and fits cleanly within the Transformer framework.

Comprehensive Experimental Setup: The paper covers several datasets and baselines, with ablation studies provided.

**Weaknesses:**

Marginal Empirical Gain

- Reported improvements are small and inconsistent, often within 0.3–0.8% of existing SOTA, and not statistically significant.

- On uniform distributions (the most common benchmark), performance even drops slightly compared to baselines.

- No clear evidence that the proposed mechanism is the cause of the gain—ablation is coarse (only “with/without” density).

 Computational Inefficiency

- Multi-scale density extraction involves multiple KNN searches per decoding step, leading to heavy runtime.

- No runtime or GPU cost comparison is provided.

- The approach thus trades a large computational burden for minor accuracy improvement.

 Weak Theoretical Support

- The claim that density modeling improves generalization remains intuitive; no theoretical or analytical justification is given.

- Lacking formal discussion on how density correlates with solution robustness or model invariance.

Poor Generalization Scope

- Despite “Generalizable” in the title, experiments are confined to TSP.

**Questions:**

- What is the computational complexity of the multi-scale density extraction?

- Why does GDaT underperform on uniform distributions? Does density modeling introduce noise when structure is homogeneous?

- Have you tested GDaT on any task beyond TSP to validate “generalizable”?

- Is there any quantitative evidence (e.g., attention map entropy, embedding variance) showing that density actually affects the attention behavior?

---

### Official Review · Reviewer_Lz1m · 2025-11-06

**Soundness:** 3
**Presentation:** 3
**Contribution:** 2
**Rating:** 4
**Confidence:** 3

**Summary:**

This paper introduces the Generalizable Density-aware Transformer (GDaT), a novel constructive solver for the Euclidean Traveling Salesman Problem (TSP). The central hypothesis is that existing Neural Combinatorial Optimization (NCO) solvers fail to generalize across both problem scales and unseen node distributions ("omni-generalization") because they neglect the regional density of unvisited nodes. To address this, GDaT proposes two core modules: (1) a multi-scale density extraction module that, at each decoding step, uses $k$-nearest neighbors and Gaussian Kernel Density Estimation (KDE) to compute density features for all unvisited nodes ; and (2) a density-aware attention module that injects these density features into the query and key vectors of a Transformer-based decoder to guide next-node selection. The authors claim that this density-aware mechanism enables GDaT to achieve state-of-the-art omni-generalization performance, which they validate through extensive experiments on synthetic and real-world TSP benchmarks.

**Strengths:**

The paper's core idea of explicitly computing and leveraging multi-scale node density features is a conceptually novel and intuitive contribution to the NCO field. The experimental evaluation is extensive, testing the model across a wide range of problem sizes (up to 10,000 nodes), four distinct synthetic distributions, and two real-world benchmarks (TSPLIB95 and National TSP). The empirical results are strong in many categories, particularly demonstrating robust performance and low variance on non-uniform distributions, as shown in Figure 3.

**Weaknesses:**

The authors' claim to be the "first NCO constructive approach... to address... omni-generalization" 1 (line 80-83) appears to be an overstatement. The task and term were previously introduced and addressed by Zhou et al. (2023), who proposed a meta-learning solution 1 (line 60, 600).2 This positioning issue extends to the baseline comparisons in Table 1 1 (line 327), which are difficult to interpret. The proposed GDaT (G) (greedy inference) is compared against baselines using a mix of powerful, non-native heuristics, such as Divide-and-Conquer (D&C), multi-rollout (G*), and 2-opt search 1 (line 325-326). This 'apples-to-oranges' comparison makes it impossible to disentangle the true architectural advantage of GDaT from the gains of external search strategies.

The paper's central hypothesis is strongly validated by its ablation study (Table 3) 1 (line 434), where removing the density module causes solution quality on large instances to collapse (e.g., 5.73% to 21.60% gap). However, the implementation of this idea introduces a fatal flaw. The model is built upon a linear attention backbone (SIL) 1 (line 368) chosen for its $O(n)$ per-step complexity 1 (line 794) 4, but the new density module re-introduces a severe $O(n \log n)$ or $O(n^2)$ bottleneck by requiring a $k$NN search 1 (line 222) 5 and KDE 1 (line 229) 6 for all $O(n)$ unvisited nodes at every decoding step. The paper's admission of "computational overhead" 1 (line 481) is a critical understatement; Table 1 1 (line 327) shows GDaT takes 2.9 hours for TSP-10000. This is orders of magnitude slower than NCO peers like GELD (3.7 minutes) and INVIT (55 seconds) and is nearly as slow as the traditional solver LKH3 (3.1-3.5 hours) while delivering a far inferior 3.09% gap. This trade-off negates the primary advantage of NCO methods (fast inference) and renders the approach non-viable.

Finally, the work's elegance is diminished by its regression to manual feature engineering. The core innovation—the multi-scale density module—is not learned end-to-end. As admitted in Appendix F.1 1 (line 741-755), the neighborhood scales $\{16, 57, 98\}$ are hand-crafted, pre-determined by a statistical analysis of the optimal solutions of the TSP-100 training set. This reliance on a priori knowledge of the training data's optimal structure questions the model's true generalization capabilities, as it may fail on new distributions that do not conform to this engineered prior.

**Questions:**

Please provide a formal computational complexity analysis (Big-O notation) for a single decoder step of GDaT, including the $k$NN search  and KDE  for all $O(n)$ unvisited nodes. How do you reconcile this $O(n \log n)$ or $O(n^2)$ cost with the $O(n)$ complexity per step of the linear attention backbone  you build upon?

Your work  claims to be the first constructive approach for omni-generalization. Could you please clarify this claim in light of prior work by Zhou et al. (2023) , which explicitly introduced the "omni-generalization" problem and proposed a meta-learning solution?

In Table 1 , your GDaT (G) method is compared against baselines using different inference strategies (e.g., D&C, G*, 2-opt). To fairly assess the architectural contribution, could you provide results for GDaT enhanced with these same strategies, or alternatively, a baseline comparison where all methods use only greedy (G) inference?

A key baseline for large-scale generalization is GELD (Xiao et al., 2025) , which reports solving TSPs up to 744k nodes.6 Given that GDaT is ~47x slower than SIL  on 10k nodes, how does GDaT's inference time and solution quality scale on instances significantly larger than 10k (e.g., 50k or 100k nodes) compared to GELD  and SIL?

---

### Meta-Review · Area_Chair_sA68 · 2026-01-05

**Summary:**

Reviewers' concerns are mainly around the following points:

W1. Overstatement on the contribution

W2. The proposed method significantly increases computational overhead but with marginal performance gain

W3. Reliance on hand-crafted features

W4. Weak Theoretical Support

W5. Evaluation scope is limited to TSP only

W6. Lacks comparison to SOTA methods

**Reviewer Concerns:**

Authors did not provide any rebuttal, so all the above concerns remain.

**Reviewer Scores:**

Since no rebuttal is provided, reviewers' opinions are unlike to change.

---

### Decision · Program_Chairs · 2026-01-26

Reject